# Development of a Novel Passive-Dynamic Custom AFO for Drop-Foot Patients: Design Principles, Manufacturing Technique, Mechanical Properties Characterization and Functional Evaluation

Paolo Caravaggi [1], Alessandro Zomparelli [2], Giulia Rogati [1,*], Massimiliano Baleani [1], Roberta Fognani [1], Franco Cevolini [3], Cristina Fanciullo [1], Arianna Cinquepalmi [1], Giada Lullini [4], Lisa Berti [1] and Alberto Leardini [1]

[1] IRCCS Istituto Ortopedico Rizzoli, 40136 Bologna, Italy; paolo.caravaggi@ior.it (P.C.); baleani@tecno.ior.it (M.B.); roberta.fognani@ior.it (R.F.); cristina.fanciullo@gmail.com (C.F.); arianna.cinquepalmi@gmail.com (A.C.); lisa.berti@ior.it (L.B.); leardini@ior.it (A.L.)
[2] CREATE Group, Section for Civil and Architectural Engineering, University of Southern Denmark (SDU), 5230 Odense, Denmark; alzo@iti.sdu.dk
[3] CRP Technology, 41126 Modena, Italy; franco.cevolini@crp-group.com
[4] IRCCS Istituto Scienze Neurologiche, 40139 Bologna, Italy; giada.lullini@isnb.it
* Correspondence: giulia.rogati@ior.it

**Abstract:** Ankle foot orthoses (AFOs) are medical devices prescribed to support the foot and ankle of drop-foot patients. Passive-dynamic AFOs (PD-AFOs) are an effective solution for less severe cases. While off-the-shelf PD-AFOs are rather inexpensive, they provide poor anatomical fit and do not account for the required patient-specific biomechanical support. Three-dimensional (3D) scanning and manufacturing technologies allow manufacturing PD-AFOs customized for the patient's anatomy and functional needs. This paper aimed to report the overall procedure for designing and manufacturing a novel, fiberglass-reinforced polyamide, custom PD-AFO. The feasibility of the proposed procedure was tested in a case study. The methodology can be divided into the following steps: (i) foot and leg scanning, (ii) 3D design, and (iii) additive manufacturing via selective laser sintering. A custom PD-AFO was designed and manufactured for a 67-year-old male drop-foot patient following paraparesis in severe discarthrosis after spine stabilization surgery. AFO mechanical properties were measured via an ad hoc setup based on a servohydraulic testing machine. The functional outcome was assessed via gait analysis in three conditions: shod (no AFO), wearing an off-the-shelf PD-AFO, and wearing the patient-specific PD-AFO. As expected, wearing the PD-AFO resulted in increased ankle dorsiflexion in the swing phase with respect to the shod condition. Sagittal rotations of the hip, knee, and ankle joints were similar across PD-AFO conditions, but the custom PD-AFO resulted in faster walking speed with respect to the off-the-shelf (walking speed: 0.91 m/s versus 0.85 m/s). Additionally, the patient scored the custom PD-AFO as more comfortable (VAS score: 9.7 vs. 7.3). While the present analysis should be extended to a larger cohort of drop-foot patients, the novel PD-AFO seems to offer a valid, custom solution for drop-foot patients not satisfied with standard orthotics.

**Keywords:** ankle foot orthosis (AFO); dynamic AFO; composite material; polyamide; fiberglass; custom; mechanical properties; drop-foot; functional evaluation; gait analysis

## 1. Introduction

Ankle foot orthoses (AFOs) are medical devices prescribed to support the foot of patients suffering from weakness of the ankle plantar/dorsiflexion muscles. This can be due to damage to the central or peripheral nervous system, such as peroneal nerve injury and spinal cord disorders. In the older population, about 20% of stroke survivors develop a drop-foot [1], whereas cerebral palsy, either hemiplegic or diplegic, is one of

the main causes in children [2]. AFOs provide stability to the ankle and prevent the foot from dropping in the swing phase of gait, thus limiting stumbling and falling injuries. Passive AFOs—those not fitted with actuators or other active devices—are classifiable into two main groups: (1) solid or rigid and (2) dynamic. Rigid AFOs are characterized by stiff shells that prevent ankle movement in any anatomical plane, whereas passive-dynamic AFOs (PD-AFOs) are more compliant in the sagittal plane and allow for some dorsi/plantarflexion movement. The latter are characterized by energy absorption/release during the stance phase of gait and are most indicated for less severe drop-foot patients. Although off-the-shelf PD-AFOs are suitable for most patients, they are sold in limited sizes and do not always address other leg and foot morphological alterations. Advancements in additive manufacturing allow obtaining custom PD-AFOs in different plastic and composite materials of virtually any shape [3]. In general, custom PD-AFOs are customized to fit the patient's foot and leg morphology. While personalization of the mechanical properties can be obtained by adjusting the calf or ventral shell bending stiffness [4–6], no clear or standard methods to establish the correct bending stiffness with respect to the patient's weight, physical impairment, and functional demand have been reported to date. The decision on the optimal AFO flexibility often relies on the orthotist experience and/or on the patient's feedback on perceived comfort, donning easiness, and subjective feeling during walking trials [7,8].

The use of hinge joints fitted with precompressed springs between the calf shell and footplate has also been proposed to provide a stiffness-controlled dynamic response to flexion/extension movements [5,9]. However, the most common designs include a posterior leaf spring (PLS) that allows the ankle joint some dorsi/plantarflexion motion. Thermoplastic, such as polypropylene, or a combination of thermoplastics and metals are the most widely used materials for PD-AFOs [10–13]. Carbon-fiber-reinforced plastics have been considered for their high mechanical properties [8,14–16]. Alternatively, fiberglass-reinforced plastics appear to be potentially suitable materials for PD-AFOs. In fact, while carbon fiber reinforcement ensures excellent strength-to-weight properties for components undergoing small deformation, fiberglass reinforcement may be more appropriate for dynamic large-deflection bending such as that occurring in PLS PD-AFOs. One of such composite materials, a fiberglass-reinforced polyamide (Windform® GT, CRP Technology, Modena, Italy), has thus far been used to manufacture light, flexible, and stiff parts for the motor and aerospace industry. In this study, it was hypothesized that this composite material could be suitable also to manufacture PD-AFOs.

This is the first study describing the overall procedure for the design and manufacturing of a PD-AFO made of fiberglass-reinforced polyamide. The feasibility and reliability of the proposed procedure were evaluated in a case study.

## 2. Materials and Methods

This section reports (i) the steps of the procedure for the design and manufacturing of the novel, custom PD-AFO and (ii) a pilot study reporting the application of the procedure in an exemplary patient suffering from drop-foot.

### 2.1. Custom PD-AFO Design and Production

As for any AFO, the present novel, custom PD-AFO must be capable of sustaining the foot during the swing phase of walking, i.e., after push-off when the foot is off the ground. The main design criteria were dictated by the need for a comfortable and light object that is easy to wear with most shoes. In general, the custom AFO was designed to fit the overall shape of the leg and foot of the patient. In addition, a specification was brought forward for the footplate to address possible foot postural alterations such as excessive foot pronation or supination. For example, in the case of severe foot pronation, the footplate can be shaped to apply some inversion to the mid- and rearfoot. The AFO features a posterior hole to allow easy donning and comfort of the calcaneus.

As stated above, AFOs must be able to sustain the foot during the swing phase of walking, thus resisting ankle joint plantarflexion moments due to the weight and the inertial forces acting on the foot and footwear (Figure 1). The cross-section of the present calf shell is dimensioned to provide enough stiffness to prevent the foot from dropping in the swing phase of gait while keeping the maximum stress of the regions subjected to the largest deformations below the material yielding strength. The calf shell is curved in the transverse plane; this provides enough rigidity and ensures rigid rotation about the axis close to that of the ankle joint. The thickness of the footplate is such that the forefoot can easily flex at the metatarsophalangeal joint at push-off while providing enough rigidity at the rear- and midfoot to prevent ankle plantarflexion. With respect to leg posture in double-leg standing, the footplate is tilted by 5 deg in dorsiflexion with respect to the calf shell; this correction ensures better foot clearance with the ground in the swing phase.

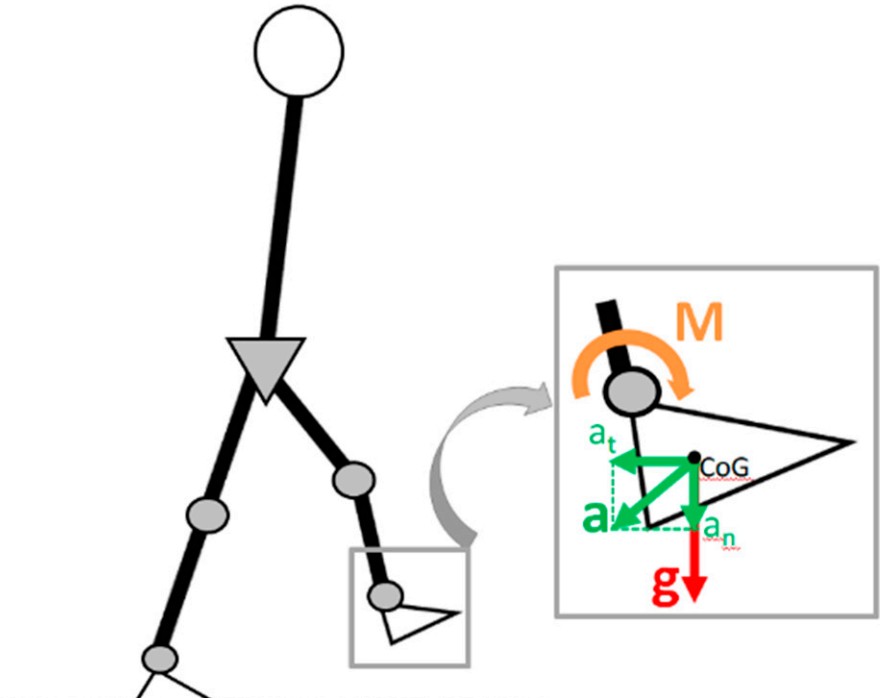

**Figure 1.** Schematic representation of the sagittal-plane components of the gravitational and inertial forces acting on the foot during the swing phase of gait. The acceleration due to the foot/leg swing (a) and to the gravity (g) applies a plantarflexion moment (M) to the ankle that must be sustained by the dorsiflexion muscles.

### 2.1.1. Foot and Leg Scanning

In order to obtain the shape and dimension of the custom AFO, a weight-bearing scan of the foot and leg [17] is performed. The subject stands on a 15 mm thick glass plate, and a Kinect depth sensor (Microsoft, Redmond, WA, USA) is manually rotated by 360 deg to obtain a full 3D representation of the foot plantar surface in weight-bearing. A scan of the leg in the same weight-bearing posture is also obtained by moving the Kinect sensor around the subject. A commercial software (Skanect version 1.8, Occipital, Inc., San Francisco (CA), USA) is used for the raw 3D point cloud data acquisition up to 21 fps. The raw 3D data are preprocessed in Skanect, and the STL files are then imported into Geomagic Control (Geomagic Control version 12; 3D Systems, Rock Hill, SC, USA) for finer editing, by removing possible noise and undesired parts. The 3D scans of the foot and leg are merged to create a full 3D representation of the weight-bearing lower limb (Figure 2), which is used to design the custom AFO.

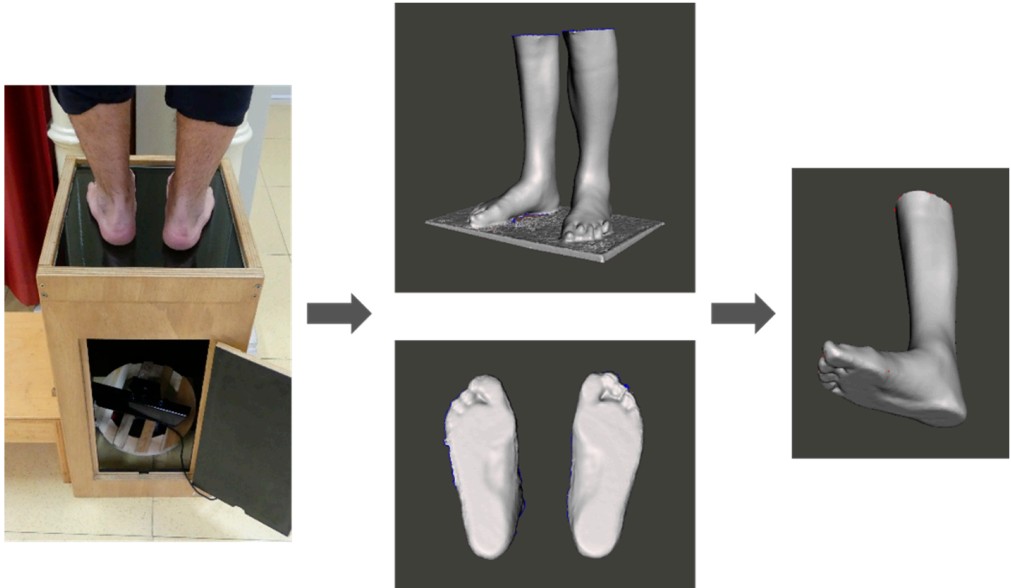

**Figure 2.** The plantar aspect of the patient's feet is 3D scanned in weight-bearing via the Kinect-based 3D scanning device. Foot plantar surface and leg geometry are acquired separately and merged into one 3D image.

### 2.1.2. 3D Design

The custom PD-AFOs are designed in Blender (www.blender.org, Blender Foundation, accessed on 10 December 2021) by matching the AFO surface to the subject's foot and leg geometry in weight-bearing (Figure 3, left). Anatomical congruence is pursued at the foot plantar surface and at the calf strap to ensure comfort and proper fixation of the AFO to the leg. Since the AFO can bend during walking, attention is taken to avoid any possible contact/sliding between AFO and foot bones that may cause discomfort and pain.

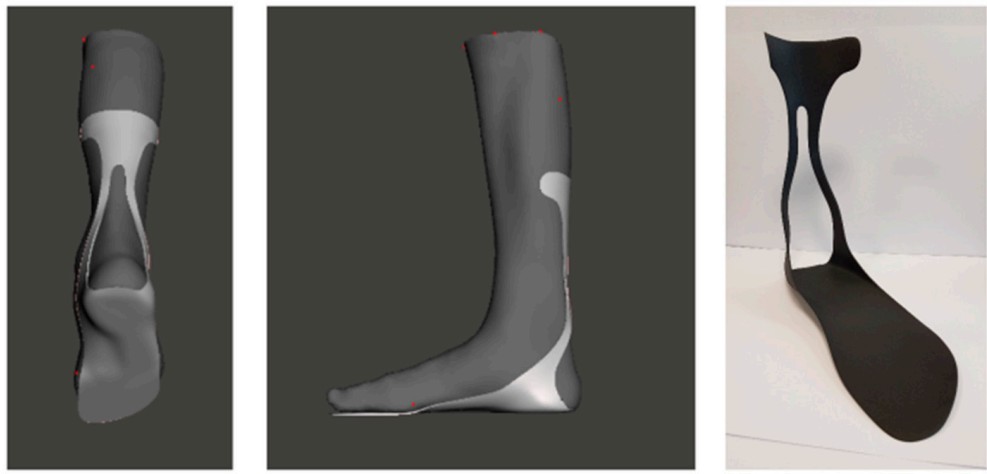

**Figure 3.** (**Left**) and (**middle**), morphology-based design of the custom AFO with respect to the 3D model of the foot and leg. (**Right**), the AFO is manufactured via selective laser sintering.

Blender is a flexible mesh modeling software that allows easy editing and postural correction of the 3D scanned limb. Some control points are manually adapted to ensure the correct alignment of the AFO to the patient's leg. The modeling process is based on Blender's Modifiers, nondestructive geometrical operations that allow a semi-implicit modeling workflow. They can be used to ensure a real-time anatomical adaptation and dimensioning of the AFO. Blender allows advanced manual and real-time control of

freeform geometries. This allows the operator to constantly refine the device through both parametric and manual editing to provide a design fully tailored to the patient's geometry. The main AFO linear dimensions and thickness are parametrically associated with the subject's foot length and with the body weight.

### 2.1.3. Additive Manufacturing

The present PD-AFOs are produced via selective laser sintering (SLS) of a fiberglass-reinforced-polyamide–based powder (Windform® GT). This is a highly elastic and flexible material (tensile strength = 56 MPa; tensile modulus = 3.3 GPa; elongation at break = 14.8%; bending strength = 88 MPa; bending modulus 3.2 MPa), has been granted skin contact safety certification, and thus it appears to be potentially suitable for this novel, custom PD-AFO.

According to the SLS process, microscopic particles of plastic polymer (Nylon) are melted by a high-power laser to form a solid three-dimensional object. The powder is initially released on a platform and laser sintered to form a 0.1 mm compact layer of material. The parts are built layer by layer, with the build platform lowering after each layer. The process takes place in a nitrogen-filled chamber to prevent oxidation (oxygen concentration < 1%). The printer preheats the powder to 10 °C below the melting point of the raw material (melting point of Windform® GT = 193 °C). The unfused powder supports the part during printing, avoiding the need for supporting structures. After printing, the build chamber needs first to slightly cool down inside the printer enclosure and subsequently outside the printer to ensure optimal mechanical properties and to reduce risks of warping. The printed parts are then removed from the build chamber, separated, and cleaned from the excess powder. A three-dimensional scan is performed to measure changes in the AFO dimensions over time. The average production time of the AFO, from the delivery of the 3D model file, is about 24 h.

### 2.2. Pilot Study

The full procedure described above was applied to an exemplary patient (age 67 years; height 175 cm; BMI = 25.6 kg/m$^2$) suffering from drop-foot due to paraparesis in severe discarthrosis after spine stabilization surgery T9-S1 and L3 osteotomy. The study was approved by the Ethical Committee of the hosting institution (No. 0016384, 23 December 2019), and the patient provided informed consent to participate in the study. The patient was selected according to the following Inclusion criteria: drop-foot due to nerve compression at the lumbosacral region of the spine; insufficiency of the ankle dorsiflexion muscles (power $\leq$ 3, according to the Medical Research Council Scale for power [18]), and compliancy to wear the custom AFO for at least 6 months. Disorders of the central nervous system, severe degenerative diseases, and/or BMI > 30 kg/m$^2$ were considered exclusion criteria. The leg and foot of the patient were 3D scanned as described in Section 2.1.1. The AFO was dimensioned and designed as described in Sections 2.1 and 2.1.2 and manufactured via selective laser sintering of fiberglass-reinforced-polyamide–based powder (Section 2.1.3). In order to measure the actual mechanical properties, the AFO was tested using an ad hoc testing setup (Section 2.2.1). The functional outcome was assessed via gait analysis (Section 2.2.2).

### 2.2.1. AFO Mechanical Properties Characterization

A setup based on an axial/torsional testing machine (Biaxial 858 Mini-Bionix, MTS System Corp., Eden Prairie, MN, USA) was used to assess the AFO stiffness by simulating the AFO motion due to the flexion/extension angular displacement of the ankle joint in the midstance phase of walking (Figure 4, left). Within the apparatus, the footplate is fixed, and the actuator axis is aligned with the ideal ankle joint axis according to anatomical proportions [19]. The AFO calf strap constrains the upper border of the calf shell on a plastic cylinder free to slide along and to rotate about the mock leg; this frictionless condition allows measuring the ideal AFO stiffness by minimizing the effects of the constraints between the AFO and leg. The actuator rod rotates under angular-displacement control up

to 15 deg in flexion, while the torque cell measures the AFO-resisting torque. The average slope of the torque/angle curve across five test repetitions is used to determine the AFO stiffness (N*m/deg).

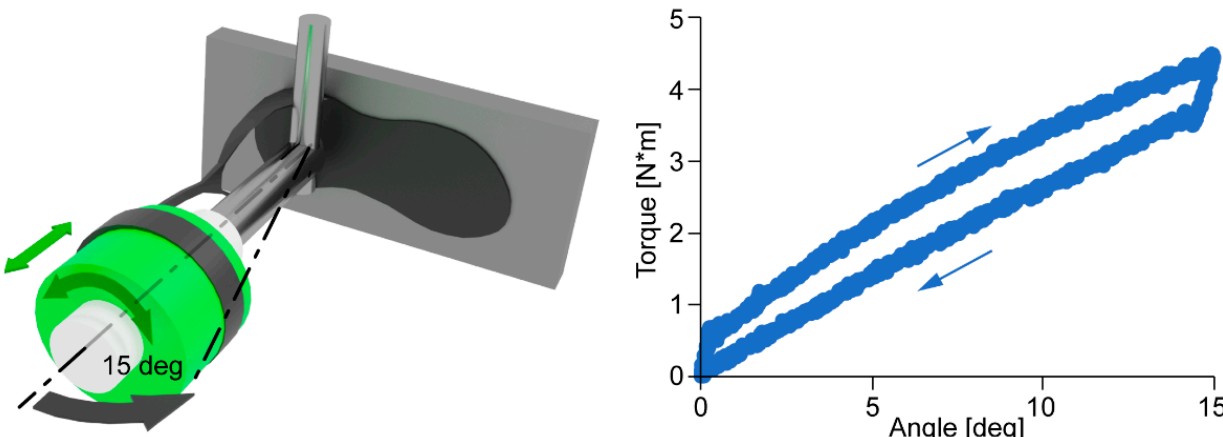

**Figure 4.** (**Left**), 3D model of the experimental setup used to measure the mechanical properties of the custom AFO. Controlled-rotations are applied to the AFO via the actuator of a servohydraulic testing machine. A torque cell measures the AFO resistance to flexion/extension rotations. (**Right**), angle/torque data for a 0-15-0 deg flexion/extension controlled displacement.

### 2.2.2. Functional and Comfort Evaluation

The present patient was asked to walk at a comfortable walking speed wearing his own shoes in the gait lab in three conditions: wearing the custom PD-AFO, wearing an off-the-shelf polyethylene AFO ("Codivilla" spring type), and without AFO (shod). The latter was used as a control to assess the kinematic and kinetic effects of the AFO conditions. Five walking trials were recorded for each condition, the order of which was randomized. The patient attended a familiarization session with the two AFOs for several minutes before undergoing gait analysis using a validated skin-marker-based protocol [20]. Accordingly, reflective markers were attached to relevant body landmarks of the upper trunk, pelvis, and lower limb. Both the affected and the contralateral legs were instrumented with markers. Rotations of the pelvis, hip, knee, and ankle joints in the three anatomical planes were calculated by a joint coordinate system [21], as recommended by the International Society of Biomechanics [22]. Ground reaction forces were sampled at 2000 Hz using two force plates embedded in the floor. Wireless sEMG sensors (Zerowire, Cometa) were placed on the tibialis anterior, gastrocnemius, rectus femoris, and biceps femoris muscles to sample muscle activity at 2000 Hz. All kinematic, kinetic, and sEMG data were normalized to gait cycle duration. Comfort of each condition was assessed via a visual analog scale (VAS) questionnaire. The VAS comprised the following scores: comfort of the footplate, comfort of the calf shell, overall comfort, and perceived support to the pushing phase.

### 3. Results

Figure 2 shows the 3D scanning of the patient's feet and legs in weight-bearing [18]. The 3D model and the laser-sintered custom AFO developed for the present patient are shown in Figure 3. Figure 4 shows the scheme of the experimental setup for the mechanical characterization of the custom AFO (Figure 4, left) and a 0-15-0 deg flexion/extension cycle (Figure 4 Right). The average stiffness of the custom AFO was 0.25 N*m/deg.

Figure 5 shows the outcome of the gait analysis. Sagittal-plane rotation (mean ± SD) of the hip (top), knee (middle), and ankle joint during normalized gait cycle rotations across five walking trials are shown for the custom AFO, Codivilla AFO, and shod conditions. Increased ankle dorsiflexion is significant in both AFO conditions. In terms of spatiotemporal parameters, the patient walked faster with the custom AFO (0.91 ± 0.04 m/s) with respect to the Codivilla (0.85 ± 0.03 m/s) and the shod (0.67 ± 0.01 m/s) conditions. The

patient scored the custom AFO as more comfortable than the Codivilla in the leg support (9.6 vs. 7.5), in the plantar aspect (9.8 vs. 3.5), and overall (9.7 vs. 7.3). The perceived push was also higher for the custom AFO (9.7 vs. 6.9).

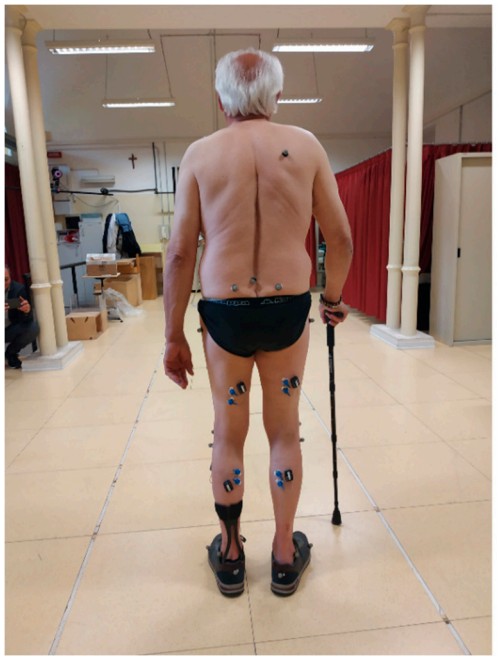
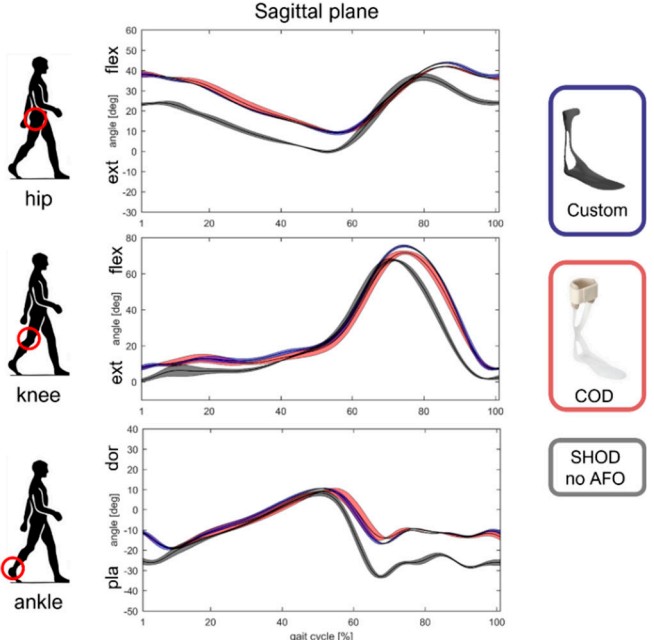

**Figure 5.** For the patient recruited in the study, sagittal-plane rotations (deg) of the hip, knee, and ankle joints during normalized gait cycle duration in the custom AFO, Codivilla (COD), and shod conditions. Joint angles are shown as mean bands (±1SD) across five walking trials.

## 4. Discussion

This study reports on the development of a new custom PD-AFO for active patients with light to mild drop-foot conditions. The PD-AFO is manufactured via selective laser sintering using a composite material that, thus far, has never been used for biomedical and orthopedic applications. The present manuscript describes the full workflow, from the patient's scanning to the 3D printing, characterization of mechanical properties, and functional evaluation. An exemplary application of the full procedure for a drop-foot patient is reported.

Designing a new PD-AFO should comprise the analysis and critical selection of a combination of optimal ergonomic and biomechanical parameters that can be addressed with a variety of shapes and materials. With this PD-AFO, the authors were willing to provide less severe drop-foot patients, who are not satisfied with standard solutions, with an aesthetically appealing, light, and comfortable device that can be worn within their shoes. The aesthetic aspect is also becoming increasingly important when choosing an orthotic device. The current design comprises three parts: a footplate with arch support matching the weight-bearing foot plantar surface of the subject, calf support shaped around the leg, and a posterior hole to allow easy donning and improve the comfort at the calcaneus. In order to improve the aesthetics, the AFO design aimed to be very simple to minimize the encumbrance and enhance the wearability of any shoe and clothing.

While the current setup requires some steps to be climbed to reach the scanning plate, which could be difficult for patients with low mobility, scanning the patient's feet and leg proved to be easy to perform. The quality of the 3D reconstruction of the foot plantar surface in weight-bearing was previously validated using a laser scanner device as the gold standard [18]. To further improve the setup, the rotating platform supporting the Kinect sensor could be driven by an electric motor to increase the scanning repeatability and thus minimize operator variability in the scanning process.

The cross-section of the calf shell, i.e., the flexible part of the PD-AFO, was dimensioned to obtain a bending stiffness consistent with the average values reported in the literature [23]. Larger sections are associated with a more rigid calf shell, as well as with larger strains and stresses in the more external regions (under the same deformation). In order to assess the actual stiffness of the calf shell of the printed AFO, this was determined by the slope of the torque/angle curve acquired with the experimental testing setup across five flexion/extension trials. While a larger sample size is required to assess the repeatability of the present experimental setup in measuring the PD-AFO stiffness, this was a highly consistent intertrial (CV < 5%). This stiffness value is rather consistent with what reported was by [23]. While the AFO stiffness was customized based on the patient's weight, future developments should investigate the criteria for the identification of the optimal mechanical properties considering other patient-specific constraints and requirements, such as the degree of functional impairment and the level of activity.

Ultimately, any orthotic device such as the current custom PD-AFO must be capable of addressing the functional deficit it was designed for. Moreover, since AFOs are usually worn several hours per day, comfort is another critical parameter that contributes to the overall evaluation. In order to objectively measure the expected improvement in walking, the present design was assessed via gait analysis in controlled laboratory conditions using a validated and widely used skin-marker-based lower limb protocol. In terms of ankle motion, wearing the PD-AFO resulted in increased dorsiflexion in the swing phase with respect to the shod condition (Figure 4). This is the primary expected outcome of the device, as the patient can confidently swing the foot by clearing the ground, thus decreasing the risk of falling due to passive foot plantarflexion. This has ensured more confidence in walking and faster walking speed. In addition, less compensation, i.e., lower range of motion in the sagittal and frontal plane, was observed at the knee and hip joints. In terms of comfort, the patient evaluated the custom PD-AFO better than the off-the-shelf AFO. In particular, the comfort of the plantar aspect and the perceived push were given higher scores.

Since the main objective of this study was to test the feasibility of the overall procedure, other than the optimization of the PD-AFO design and its mechanical properties, the latter aspects will require further analyses and investigations. A parametrization of the main dimensions should be sought to provide the custom PD-AFO with morphology-based shape and mechanical properties fitting the deficit and the functional demand of each patient. In addition, while the outcome of the functional evaluation is encouraging, a larger sample of drop-foot patients should be recruited in future studies to include a larger variety of lower limb shapes, degrees of ankle deficit, and foot postural alterations. In terms of functional evaluation, while walking is possibly the most common daily motor task, other motor tasks such as slow running and stair ascending and descending could be tested in future endeavors.

## 5. Conclusions

The current procedure allows for a comprehensive evaluation of the custom PD-AFO mechanical properties and the effects on gait. For the exemplary drop-foot patient recruited here, the custom PD-AFO was more comfortable than a standard orthosis, supported the foot in the swing phase of gait, and allowed for faster walking speed with respect to the no-AFO condition. While the present functional outcome will require confirmation in a larger cohort of drop-foot patients, the procedure appears suitable to further improve the PD-AFO design in terms of biomechanics and subject-specific functional limitations and demand.

**Author Contributions:** Conceptualization, P.C. and M.B.; methodology, P.C., M.B. and G.R.; software, A.Z. and G.R.; validation, P.C. and M.B.; experimental investigation, R.F., M.B., P.C. and G.R.; resources, F.C., A.L. and M.B.; data curation, M.B., R.F., P.C., G.R., A.Z., C.F. and A.C.; writing—original draft preparation, P.C. and M.B., writing, review and editing, P.C., G.R., A.Z., A.L., G.L., L.B. and M.B.; supervision, P.C., M.B. and A.L.; project administration, A.L.; funding acquisition, A.L., P.C. and M.B. All authors have read and agreed to the published version of the manuscript.

**Funding:** This work was partially supported by the Italian Ministry of Health 5 × 1000.

**Institutional Review Board Statement:** The study has been approved by the Ethical Committee of the hosting institution (No. 0016384, 23 December 2019).

**Informed Consent Statement:** The patient recruited for the pilot study reported here provided informed consent to participate in the study.

**Data Availability Statement:** All experimental data are available on request.

**Acknowledgments:** The authors wish to thank Paolo Erani and Maurizio Ortolani for the technical support and Luigi Lena for the artwork.

**Conflicts of Interest:** CRP technology owns the material used for the custom PD-AFO.

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
