# Peer review of "Development of a Novel Passive-Dynamic Custom AFO for Drop-Foot Patients: Design Principles, Manufacturing Technique, Mechanical Properties Characterization and Functional Evaluation"

_applsci, doi:10.3390/app12094721_

Round 1
Reviewer 1 Report
The actual conclusion section of the article is short although it gets right to the point about the procedures they followed for the design. The discussion section goes into more detail about the objectives and backs it up with data results. Some of the information in the discussion section should be in the conclusion so the article ends with a bit more information.
Author Response
General reply to both reviewers:
The authors would like to thank both reviewers for their comments and contributions to improving the manuscript. One of the major comments was to improve the general layout and organization of the manuscript as to make the aims clearer. Thus, in this revised version, the Abstract section and the Methods have been fully revised. In particular, the Methods have been divided in a procedure/workflow paragraph, and in a “Pilot Study” paragraph which shows the application of the procedure in a clinical case of drop-foot.
All the changes from the original submission have been highlighted in yellow in this revision.
Reply tor reviewer#1:
We thank the reviewer for the nice comments. The conclusion section has been slightly revised as suggested by the reviewer.
Reviewer 2 Report
This manuscript entitled “Development of a novel passive dynamic custom AFO for drop-foot patients: design principles, manufacturing technique, mechanical properties characterization and functional evaluation” examines the overall procedure from the 3D scanning to the design, the mechanical properties characterization and the functional evaluation via gait analysis. In the reviewer's opinion, the most prominent highlight of the current research is the method part, which uses the foot and leg scan to custom the footwear, then via gait analysis to evaluate the function of different shoes, offering a more comprehensive analysis over the novel custom PD-AOF on the foot of patients suffering from weakness of the ankle plantar/dorsiflexor muscles. However, the reviewer believes that the authors are simply described the whole part of the method, such as L179-181: The present patient was asked to walk at a comfortable walking speed in the gait lab in the three conditions: wearing the custom AFO, wearing a standard Codivilla spring, and shod with no AFO, which lack of detailed description. The reviewer is very worried about this because it directly affects the quality of the article and readers' understanding of the article. At the same time, whether the patient was selected to be representative which is also the concern of the reviewer.
Specific comments are shown below:
Abstract:
The reviewer suggested that the description of the abstract part is not clear. The results described in this abstract are insufficient, and the conclusion of this paper is not clear. Therefore, the reviewer suggested the author revised the abstract. Meanwhile, the reviewer was confused about the purpose of the manuscript, just to show us AFO’s design principles, manufacturing technique, 3 mechanical properties characterization and functional evaluation?
Introduction:
This manuscript focused on a new custom AFO, but there are few studies covering this new custom POA as well as a few supporting literature. For example, “A composite material, fiberglass-reinforced polyamide based (Windform® GT), has thus far been used to manufacture light, flexible and stiff parts for the motor industry and other demanding sectors such as drones and aerospace. Due to its favorable mechanical properties, this composite material has been considered suitable for a light, flexible, resistant to impact and durable custom PD-AFO” Are there specific references to support this? The reviewer suggests the author should not be too subjective, but to be objective. Please revised, so that the readers can have a clearer understanding of the background of this research.
Methods:
Lines 72-78: are there prior literature available on the selection criteria for subjects or are there medically strict criteria for subjects with pain? Reviewers suggested that is more rigorous to clarify the criteria for the patient.
Line 179-181: the experiment phase is not clearly described. Whether the patient will not adapt to the PD-AFO of a particular material at first exposure, and will this affect the final results? The reviewer suggested that the author indicates the preparation and the specific operation of experimental control to increase the rigor of this paper.
2.6. functional and comfort evaluation: experiment protocol seems important in an experimental article, but the reviewer considered that the part of experiment control in this manuscript is unorganized. A total of how many sets of successful trials were extracted? And which legs’ ground reaction force was interested in the research (left or right or both legs)? What’s the experimental order of the three different condition shoes, is it randomly selected or is it prescribed by the experimenter? Please describe in detail in the experiment protocol section.
Results:
Line 175-177: How to use the finite element analysis, the viewer suggested the author descript in detail, to facilitate future generations to repeat this research?
Discussion:
As the author mentioned in the abstract part “this paper is reporting the overall procedure from the 3D scanning to the design, the mechanical properties characterization and the functional evaluation via gait analysis” (Lines 14-16). However, the reviewer considers that the most important is the function of the novel custom PD-AOF? The purpose is not clear and the logic is not clear.
Some recently studies could be added in the discussion, such as:
The Categories of AFO and Its Effect on Patients With Foot Impair: A Systemic Review. Physical Activity and Health, 1(1), 8–16. DOI: http://doi.org/10.5334/paah.3
Evaluating function in the hallux valgus foot following a 12-week minimalist footwear intervention: A pilot computational analysis. J Biomech. 2022;132:110941. doi: 10.1016/j.jbiomech.2022.110941.
Anterior or Posterior Ankle Foot Orthoses for Ankle Spasticity: Which One Is Better? Brain Sci. 2022, 12, 454. https://doi.org/10.3390/brainsci12040454
Line 289: As the limitations mentioned in the author’s discussion, there are big problems in the current research method, which is also the concern of the reviewer.
Author Response
General reply to both reviewers:
The authors would like to thank both reviewers for their comments and contributions to improving the manuscript. One of the major comments was to improve the general layout and organization of the manuscript as to make the aims clearer. Thus, in this revised version, the Abstract section and the Methods have been fully revised. In particular, the Methods have been divided in a procedure/workflow paragraph, and in a “Pilot Study” paragraph which shows the application of the procedure in a clinical case of drop-foot.
All the changes from the original submission have been highlighted in yellow in this revision.
Reviewer#2
Comments and Suggestions for Authors
This manuscript entitled “Development of a novel passive dynamic custom AFO for drop-foot patients: design principles, manufacturing technique, mechanical properties characterization and functional evaluation” examines the overall procedure from the 3D scanning to the design, the mechanical properties characterization and the functional evaluation via gait analysis.
In the reviewer's opinion, the most prominent highlight of the current research is the method part, which uses the foot and leg scan to custom the footwear, then via gait analysis to evaluate the function of different shoes, offering a more comprehensive analysis over the novel custom PD-AOF on the foot of patients suffering from weakness of the ankle plantar/dorsiflexor muscles. However, the reviewer believes that the authors are simply described the whole part of the method, such as L179-181: The present patient was asked to walk at a comfortable walking speed in the gait lab in the three conditions: wearing the custom AFO, wearing a standard Codivilla spring, and shod with no AFO, which lack of detailed description. The reviewer is very worried about this because it directly affects the quality of the article and readers' understanding of the article. At the same time, whether the patient was selected to be representative which is also the concern of the reviewer.
The authors fully agree with the reviewer on that the paper is mainly Methodological. The main purpose was in fact to report all the procedural steps that were implemented to design the novel custom PD-AFO and to test it. The detailed description of each step provides the reader with all the necessary information to show the feasibility of the procedure and to optimize or adapt it to his/her specific application.
In order to address the reviewer’s major comment, more details on the functional evaluation step have now been added to the Methods. In terms of the selected patient, the authors understand the choice may appear a little arbitrary but he was the very first patient testing the new fibreglass-reinforced polyamide PD-AFO while the recruitment process is still ongoing. No biased criteria were chosen to select this patient. The pilot study allows to show the potential functional impact of customised PD-AFOs. Reporting the functional outcome for the whole drop-foot population that is being recruited will be sought in a future study.
Specific comments are shown below:
Abstract:
The reviewer suggested that the description of the abstract part is not clear. The results described in this abstract are insufficient, and the conclusion of this paper is not clear. Therefore, the reviewer suggested the author revised the abstract. Meanwhile, the reviewer was confused about the purpose of the manuscript, just to show us AFO’s design principles, manufacturing technique, 3 mechanical properties characterization and functional evaluation?
The authors agree with the reviewer on that the abstract could be better tailored to the manuscript actual content. To address the reviewer’s comment, the abstract has now been fully revised and some results are reported in the relevant section. For the complexity of the design and of the experimental procedure, this paper only aimed at reporting the details of the whole methodology. Each of these steps, and in particular the mechanical and functional characterizations, may be expanded to a full study reporting the results for the whole population – the recruitment and data analysis of which is still ongoing. It was therefore decided to report functional and mechanical outcomes for an exemplary drop-foot case only (the new “pilot study” section). While the functional outcome and the kinematic corrections due to the AFO are largely as predicted, the present results should be considered as preliminary just to show the whole procedure. As stated above, we hope the reviewer will appreciate that in this revised version, the Abstract section and the Methods have been fully revised. In particular, the Methods have been divided in a procedure/workflow paragraph, and in a “Pilot Study” paragraph which shows the application of the procedure in a clinical case of drop-foot. This new subdivision of the paragraphs in the Methods appears to be more consistent with the aims of the study as described in the revised Abstract.
Introduction:
This manuscript focused on a new custom AFO, but there are few studies covering this new custom POA as well as a few supporting literature. For example, “A composite material, fiberglass-reinforced polyamide based (Windform® GT), has thus far been used to manufacture light, flexible and stiff parts for the motor industry and other demanding sectors such as drones and aerospace. Due to its favorable mechanical properties, this composite material has been considered suitable for a light, flexible, resistant to impact and durable custom PD-AFO” Are there specific references to support this? The reviewer suggests the author should not be too subjective, but to be objective. Please revised, so that the readers can have a clearer understanding of the background of this research.
This is an original paper reporting the procedure and together with some preliminary mechanical and functional data on this novel fibreglass reinforced polyamide custom PD-AFO, therefore this has now been better specified in the Introduction. The authors agree on that the description of the material in the Introduction and also in the Methods sections is a little biased, thus these parts have now been modified to address the reviewer’s sensible point.
Methods:
Lines 72-78: are there prior literature available on the selection criteria for subjects or are there medically strict criteria for subjects with pain? Reviewers suggested that is more rigorous to clarify the criteria for the patient.
Subjects were included in the study according to the inclusion criteria approved by the Ethical Committee of the host Institution. Inclusion and exclusion criteria have now been better described in paragraph 2.2 of the Methods.
Line 179-181: the experiment phase is not clearly described. Whether the patient will not adapt to the PD-AFO of a particular material at first exposure, and will this affect the final results? The reviewer suggested that the author indicates the preparation and the specific operation of experimental control to increase the rigor of this paper.
Correct, the patient attended a familiarization session with the two AFO conditions for several minutes before undergoing the gait analysis. The exemplary patient reported here was already familiar with wearing AFOs, but not with the material and the design of the present custom-AFO. The testing order of the three conditions was randomized, this has now been specified in the Methods.
2.6. functional and comfort evaluation: experiment protocol seems important in an experimental article, but the reviewer considered that the part of experiment control in this manuscript is unorganized. A total of how many sets of successful trials were extracted? And which legs’ ground reaction force was interested in the research (left or right or both legs)? What’s the experimental order of the three different condition shoes, is it randomly selected or is it prescribed by the experimenter? Please describe in detail in the experiment protocol section.
The authors agree with the reviewer on this point and more details on the functional experimental conditions have now been provided in the revised manuscript, such as the number of walking trials and the randomization routine. Both legs were instrumented with skin-markers and sEMG sensors. The shod condition was used here as control to assess the kinematic and kinetic effects of the two AFO conditions.
Results:
Line 175-177: How to use the finite element analysis, the viewer suggested the author descript in detail, to facilitate future generations to repeat this research?
The boundary conditions of the FEA replicated those of the mechanical setup assessing the stiffness properties. The number and type of elements is not easy to establish as it depends on the software used and on the computer specifications running it. A full new paragraph would be required to describe thoroughly the FEA. Therefore, since the authors do not believe this part to be highly critical to the aims of the present paper, the FEA section has now been removed from the Methods along with the relevant comment in the Discussion.
Discussion:
As the author mentioned in the abstract part “this paper is reporting the overall procedure from the 3D scanning to the design, the mechanical properties characterization and the functional evaluation via gait analysis” (Lines 14-16). However, the reviewer considers that the most important is the function of the novel custom PD-AOF? The purpose is not clear and the logic is not clear.
The authors do apologize if the aims were not fully clear in the original submission. The abstract has now been fully revised and we hope the main purpose of this paper is now clearer. In terms of functional evaluation, the relevant paragraph has now been expanded with additional details on the methodology. This paper is mainly methodological and more functional data will be provided following recruitment and analysis of a larger cohort of drop-foot patients (ongoing).
Some recently studies could be added in the discussion, such as:
The Categories of AFO and Its Effect on Patients With Foot Impair: A Systemic Review. Physical Activity and Health, 1(1), 8–16. DOI: http://doi.org/10.5334/paah.3
Evaluating function in the hallux valgus foot following a 12-week minimalist footwear intervention: A pilot computational analysis. J Biomech. 2022;132:110941. doi: 10.1016/j.jbiomech.2022.110941.
Anterior or Posterior Ankle Foot Orthoses for Ankle Spasticity: Which One Is Better? Brain Sci. 2022, 12, 454. https://doi.org/10.3390/brainsci12040454
We thank the reviewer for the advice.
Line 289: As the limitations mentioned in the author’s discussion, there are big problems in the current research method, which is also the concern of the reviewer.
In the Discussion the authors stated that : “Since the main objective of this study was to test the feasibility of the overall procedure, other than the optimization of the AFO design and of its mechanical properties, the latter aspects will require further analyses and investigations.”
As stated also in the revised abstract, this paper was meant to be “Methodological” and only aimed at presenting the overall procedure whereas future studies will focus thoroughly on the functional and mechanical characterization of this novel custom AFO in a larger cohort of drop-foot patients.
Round 2
Reviewer 2 Report
The authors have well addressed my questions, it was an interesting study, suitable to journal's topic, So I recommend to accept.